# Status of G2HDM with right handed neutrino coupling in the light of $b \to c\tau\nu$ anomalies

**Nilakshi Das**[a] **Amit Adhikary**[b] **Rupak Dutta**[c]

[a]*Indian Institute of Technology Gandhinagar, Department of Physics, Gujarat 382355, India*
[b]*Aix Marseille Univ, Université de Toulon, CNRS, CPT, IPhU, Marseille, France*
[c]*National institute of Technology Silchar 788010, Silchar, India*

  *E-mail:* nilakshi.das@iitgn.ac.in, amit.adhikary@cpt.univ-mrs.fr,
  rupak@phy.nits.ac.in

ABSTRACT: Recent experimental measurements of several observables in semileptonic B meson decays have pointed towards the possibility of new physics. The LHCb collaboration has reported a significant deviation, exceeding $3.2\sigma$, in the combined measurement of the ratio of branching ratios $R(D) - R(D^\star)$ from the predictions of the standard model. Furthermore, other observables, such as $R_{J/\psi}$, $P_\tau^{D^\star}$, $F_L^{D^\star}$, and $R_{\Lambda_c}$ in the $b \to c\ell\nu$ transition, have also exhibited noticeable deviations from the standard model predictions. Motivated by these anomalies in the $b \to c\tau\nu$ transitions, we perform a log-likelihood fit incorporating new physics coming from right-handed neutrino couplings and explored the implications of a charged Higgs boson within a generic two Higgs doublet model (G2HDM). Our comprehensive analysis, focused on the $\tau\nu$ and $b\tau\nu$ final states, was performed using the High Luminosity run of the Large Hadron Collider (HL-LHC). We demonstrate that HL-LHC has the sensitivity to exclude the remaining allowed region in G2HDM model in explaining these anomalies with charged Higgs boson coupled to right-handed neutrino.

# 1 Introduction

The standard model (SM) of particle physics stands as a robust and extensively tested theoretical framework that has successfully described a wide range of experimental observations. However, several phenomena and experimental measurements suggest that the SM is an incomplete description of nature. Various observations such as neutrino oscillations, the existence of dark matter, the imbalance between matter and antimatter in the universe, and various flavor anomalies in particle decays collectively highlight the necessity for further research and investigations aimed at advancing our understanding of the fundamental laws of nature.

Indications of physics beyond the Standard Model (BSM) have emerged in various aspects, notably in the field of $B$ meson decays. Recent measurements from BaBar [1, 2], Belle [3–5], and LHCb [6–9] collaborations have unveiled significant deviations in the observables $\mathcal{R}_D$ and $\mathcal{R}_{D^\star}$ from the SM predictions. The Heavy Flavor Averaging Group (HFLAV) [10] indicates a tension of $1.6\sigma$ and $2.5\sigma$ between the measured values of $\mathcal{R}_D$ and $\mathcal{R}_{D^\star}$ and their respective SM expectations. Moreover, the combined world average exhibits a deviation of $3.31\sigma$ [10] from the SM prediction, accompanied by a correlation of $-0.39$ between $\mathcal{R}_D - \mathcal{R}_{D^\star}$. The LHCb collaboration has also measured the ratio of branching fractions $R_{J/\psi}$ in $B_c \rightarrow J/\psi\ell\bar\nu$ decays and observed a deviation of around $1.9\sigma$ from the SM prediction [11, 12]. Similarly, LHCb collaboration reported measurement of the semileptonic decay process $\Lambda_b \rightarrow \Lambda_c^+ \tau^- \bar\nu$, yielding a significant signal with a significance of $1.8\sigma$ [12, 13]. The measurement provides a determination of the LFUV ratio, $R_{\Lambda_c} = 0.242 \pm 0.026\,(\text{stat.}) \pm 0.071\,(\text{syst.})$. Apart from the LFUV observables, the $\tau$ polarization asymmetry, $P_\tau^{D^\star}$ [14, 15], and the longitudinal polarization fraction of $D^\star$ meson, $F_L^{D^\star}$ [16], are within $1\sigma$ of the SM expectation [12].

Several model-dependent analysis including charged scalar attempted to explain these observed anomalies. Previous studies [17–38], have shown that the charged Higgs boson ($H^\pm$) in type-III 2HDM with left-handed neutrinos can simultaneously explain the anomalies observed in $R_D$ and $R_{D^\star}$. Our work, in this regard, is significantly different from the existing ones. We aim to explore the new physics parameter space and investigate the possibility of the charged Higgs boson with right handed neutrinos to provide a consistent explanation for the observed anomalies in various LFUV observables. To this end, we start from a model-independent fit by considering the experimental measurement of all the flavor observables, namely $R_D$, $R_{D^*}$, $P_\tau^{D^\star}$, $R_{J/\psi}$, $F_L^{D^*}$ and $R_{\Lambda_c}$ in the presence of scalar new physics (NP) interaction involving right-handed neutrino. We also adopt the latest bound for the branching fraction, $\mathcal{B}(B_c \rightarrow \tau\nu) \leq 63\%$ [39] to further constrain the NP parameter space. Furthermore, we investigate the type-III Two Higgs Doublet Model (2HDM) with generic flavor structure [40, 41],

as a simple extension of the SM to explain the observed anomalies in LFUV observables.

This paper is organized as follows: The theoretical formalism, along with the relevant formulae for the LFUV observables, and the results obtained from our model-independent analyses, are presented in Section 2. In Section 3, we start with the generic 2HDM and discuss the theoretical framework along with the features of the model that are pertinent for our analysis. The collider analysis of the charged Higgs boson $(H^{\pm})$ is explored in section 4. Furthermore, we examine whether the charged Higgs with right-handed neutrino Yukawa couplings can be a possible explanation for the anomalies observed in flavor observables, in Section 5. Finally, we summarize our findings and conclude in section 6.

## 2    Scalar coupling sensitivity to the flavor observables

The effective Lagrangian for the $b \to c\ell\nu$ quark-level transition in the presence of scalar NP interactions involving right handed neutrino can be expressed as [42, 43]:

$$\mathcal{L}_{eff} = -\frac{4G_F}{\sqrt{2}}|V_{cb}|\left[(\bar{c}_L\gamma^\mu b_L)(\bar{l}_L\gamma_\mu\nu_L) + C_{LR}^S\bar{l}_L\,\nu_R\,\bar{c}_L\,b_R + C_{RR}^S\bar{l}_L\,\nu_R\,\bar{c}_R\,b_L\right] + \text{h.c.} \qquad (2.1)$$

where $G_F$ represents Fermi coupling constant and $|V_{cb}|$ represents the Cabibbo-Kobayashi-Maskawa (CKM) matrix element. The NP Wilson coefficients (WC), denoted as $C_{LR}^S$ and $C_{RR}^S$, correspond to the scalar couplings involving right-handed neutrinos. In the presence of such NP, the flavor observables $R_D$, $R_{D^\star}$, $P_\tau^{D^\star}$, $F_L^{D^\star}$, $R_{J/\psi}$, $R_{\Lambda_c}$ and $\mathcal{B}(B_c \to \tau\bar{\nu})$, at the bottom quark mass scale $m_b$, can be expressed as follows [44–47]. For our analysis, we assume

the mass of the right handed neutrino to be negligibly small:

$$R_D \simeq R_D^{\text{SM}} \left\{ 1 + 1.01 \ |C_{RR}^S + C_{LR}^S|^2 \right\},$$

$$R_{D^\star} \simeq R_{D^\star}^{\text{SM}} \left\{ 1 + 0.04 \ |C_{RR}^S - C_{LR}^S|^2 \right\},$$

$$P_\tau^{D^\star} \simeq \frac{R_{D^\star}^{\text{SM}}}{R_{D^\star}} P_{\tau,\text{SM}}^{D^\star} \left\{ 1 + 0.07 \ |C_{RR}^S - C_{LR}^S|^2 \right\},$$

$$F_L^{D^\star} \simeq \frac{R_{D^\star}^{\text{SM}}}{R_{D^\star}} F_{L,\text{SM}}^{D^\star} \left\{ 1 + 0.08 \ |C_{RR}^S - C_{LR}^S|^2 \right\}, \tag{2.2}$$

$$R_{J/\psi} \simeq R_{J/\psi}^{\text{SM}} \left\{ 1 + 0.04 \ |C_{RR}^S - C_{LR}^S|^2 \right\},$$

$$R_{\Lambda_c} \simeq R_{\Lambda_c}^{\text{SM}} \left\{ 1 + 0.53 \ \text{Re}[C_{RR}^S C_{LR}^{S\star}] + 0.33 \ (|C_{RR}^S|^2 + |C_{LR}^S|^2) \right\},$$

$$\mathcal{B}(B_c \to \tau\bar{\nu}) \simeq \mathcal{B}(B_c \to \tau\bar{\nu})_{\text{SM}} \left\{ 1 + |4.35(C_{RR}^S - C_{LR}^S)|^2 \right\}$$

Computation of all the observables in Eq. 2.2 involves the utilization of specific form factors. For $R_D$, $R_{J/\psi}$ and $R_{\Lambda_c}$, we employ the lattice QCD form factors of Ref. [48], Ref. [49] and Ref. [50], respectively. To calculate the observables $R_D^\star$, $P_\tau^{D^\star}$, and $F_L^{D^\star}$, we utilize the Heavy Quark Effective Theory (HQET) form factors of Ref. [51]. All the input parameters pertinent for our analysis are presented in Appendix B. We would like to mention that the uncertainties associated with the mass parameters and decay lifetimes of the hadrons are not taken into consideration in the analysis. However, we rigorously account for the uncertainties arising from the input parameters pertaining to the form factors and the CKM matrix element $|V_{cb}|$.

To explore the NP parameter space, we perform a $\chi^2$ fit by considering total of six measurements, namely $R_D$, $R_{D^\star}$, $P_\tau^{D^\star}$, $F_L^{D^\star}$, $R_{J/\psi}$, and $R_{\Lambda_c}$. We define the $\chi^2$ fit function as follows:

$$\chi^2 \equiv \sum_{i,j} (O^{\text{theory}} - O^{\text{exp}})_i \ \text{Cov}_{ij}^{-1} \ (O^{\text{theory}} - O^{\text{exp}})_j. \tag{2.3}$$

To perform the $\chi^2$ fit and determine the best-fit value of the scalar coupling, we utilize the `iminuit` package [52, 53]. In this analysis, we minimize a negative log-likelihood function using a multivariate Gaussian probability density function. During the fitting procedure, we account for the correlation of $-0.39$ [10] between the observables $R_D$ and $R_{D^\star}$. We assume the NP couplings to be real and varied them within the range of $(-1, +1)$. In our chi-square

analysis, we examine one NP coupling at a time and find that both couplings yield the same best-fit value. The best fit value is

$$C_{RR}^S/C_{LR}^S = -0.48 \pm 0.08.$$

The corresponding best fit value and the $1\sigma$ allowed range of each flavor observables are summarized in Table 1. It is worth mentioning that the best fit value and the $1\sigma$ allowed range of $\mathcal{B}(B_c \to \tau\nu)$ obtained with scalar NP coupling satisfies the latest bound of $\mathcal{B}(B_c \to \tau\nu) \leq 63\%$.

| WC | $R_D$ | $R_D^*$ | $R_{J/\psi}$ | $R_{\Lambda_c}$ | $P_\tau^{D^*}$ | $F_L^{D^*}$ | $\mathcal{B}(B_c \to \tau\nu)$ (%) |
|---|---|---|---|---|---|---|---|
| Best fit | 0.367 | 0.256 | 0.260 | 0.349 | $-0.500$ | 0.468 | 11.79 |
| ( $1\sigma$  allowed range ) | ( 0.327, 0.407 ) | ( 0.237, 0.276 ) | ( -0.013, 0.533 ) | ( 0.239, 0.458 ) | ( -1.304, 0.303 ) | ( 0.392, 0.544 ) | ( 8.86, 15.25 ) |

**Table 1**: *The best fit value and the corresponding $1\sigma$  allowed range of $R_D$, $R_{D^*}$, $R_{J/\psi}$, $R_{\Lambda_c}$, $P_\tau^{D^*}$ and $F_L^{D^*}$ with $C_{RR}^S/C_{LR}^S$ NP coupling.*

In Fig. 1, we present the implications of our fit result in various 2D planes of flavor observables, namely, $R_D - R_{D^\star}$, $P_\tau^{D^\star} - F_L^{D^\star}$, $R_{J/\psi} - R_{\Lambda_c}$ planes. We show the SM results and experimental central value with black and blue $\star$ symbol, respectively. The experimental uncertainty bands at $1\sigma$, $2\sigma$ and $3\sigma$ level are shown as elliptical contours with solid, dashed and dotted blue colored contours, respectively. The best fit value of each observables is represented by "$\checkmark$" symbol. The red shaded region around it corresponds to the $1\sigma$ band, coming from the experimental and theoretical uncertainties of the flavor observables.

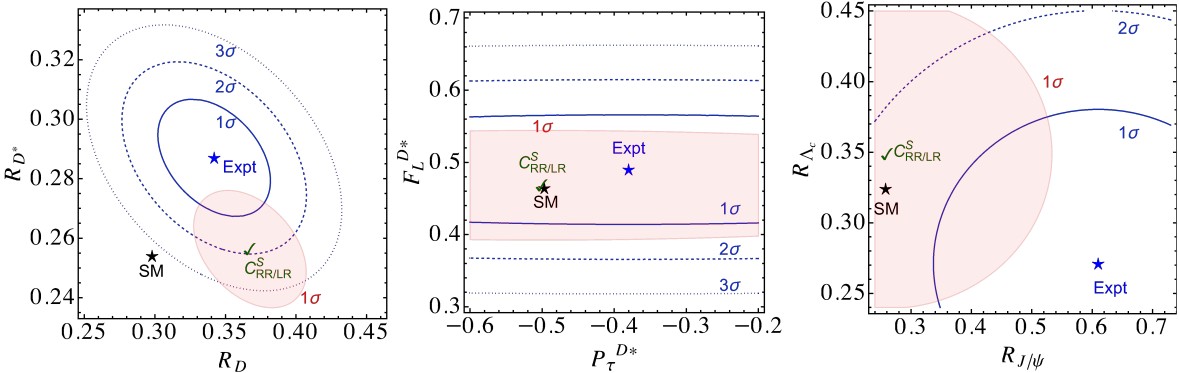

**Figure 1**: *The best fit value and the corresponding $1\sigma$ region of the flavor observables in the presence of $C_{RR}^S/C_{LR}^S$ NP coupling are shown in the $R_D - R_{D^*}$, $P_\tau^{D^\star} - F_L^{D^\star}$ and $R_{J/\psi} - R_{\Lambda_c}$ planes. The blue contours represent experimental measurement at different $\sigma$ level, whereas the SM prediction and experimental central value are denoted by black and blue star, respectively.*

As mentioned earlier in section 1, a charged Higgs in a generic 2HDM can still explain the $R_D - R_{D^\star}$ anomaly with left-handed scalar neutrino couplings. We further investigate this possibility for right-handed neutrino couplings in the subsequent sections.

## 3  Sensitivity of the G2HDM model

In our analysis, we examine the charged Higgs boson within the framework of a generic Two Higgs Doublet Model (G2HDM), where both Higgs doublets interact with up-type and down-type quarks. The charged Higgs ($H^\pm$) interaction can be expressed as follows [28]:

$$\mathcal{H} = \frac{\tilde{y}_{bc}\,\tilde{y}_{\tau\nu}}{m_{H^\pm}^2}(\bar{c}_R b_L)(\bar{\tau}_L \nu_{\tau R}) \tag{3.1}$$

The Yukawa couplings associated with the down-type Higgs doublets, $\tilde{y}_{bc}^d$, is constrained by the $B_s - \bar{B}_s$ mixing [41]. Thereby we concentrate on the up-type Yukawa, $\tilde{y}_{bc}^u$ interaction or the less constrained coupling $C_{RR}^S$. The free parameters of interest in this analysis are $\tilde{y}_{bc}^u$ (or simply denoted as $\tilde{y}_{bc}$) and $\tilde{y}_{\tau\nu}$. Our focus lies on the $b \to c\tau\bar{\nu}$ transition, mediated by a charged Higgs. This transition can be described by an effective Hamiltonian, which can be parameterized as follows:

$$\mathcal{H}_{\text{eff}} = 2\sqrt{2}G_F V_{cb}[(\bar{c}_L\gamma_\mu b_L)(\bar{\tau}_L\gamma^\mu\nu_{\tau L}) + C_{RR}^S(\bar{c}_R b_L)(\bar{\tau}_L\nu_{\tau R})]. \tag{3.2}$$

In this context, the scalar coupling $C_{RR}^S$ is defined at the scale of the charged Higgs mass ($m_{H^\pm}$) as follows:

$$C_{RR}^S = \frac{\tilde{y}_{bc}\,\tilde{y}_{\tau\nu}}{2\sqrt{2}G_F V_{cb} m_{H^\pm}^2}. \tag{3.3}$$

It is important to note that the value of $C_{RR}^S$ used in the definition of flavor observables in Eq. 2.2 is defined at the scale of the bottom quark mass ($m_b$). Hence, we employ the renormalization group equation (RGE) running of $C_{RR}^S$ from the charged Higgs mass scale ($m_{H^\pm}$) to the $m_b$ scale, following the prescribed procedure outlined in [54].

We do a collider analysis focusing on the final states $\tau\nu$ and $b\tau\nu$ in the following sections. These results will subsequently be utilized to impose constraints on the Yukawa coupling under investigation.

## 4  Collider analysis

To address the simultaneous explanation of the $R_D - R_{D^\star}$ anomaly, the collider prospects of a charged Higgs boson in the $\tau\nu$ final state have been examined in Ref. [35, 36]. It has been demonstrated that the inclusion of an extra b-tagged jet can further enhance the search potential [36]. Previous searches at the LHC have placed constraints on the decay of a charged Higgs boson into a hadronic $\tau$ ($\tau_h$), primarily through $W' \to \tau\nu$ searches [55, 56]. However, these searches still allow for a low-mass range of the charged Higgs boson, with $m_{H^\pm} \leq 400$ GeV [36].

In this study, we analyze both the $pp \to H^\pm \to \tau_h\nu$ (b-veto category) and $pp \to bH^\pm \to b\tau_h\nu$ (b-tag category) processes, at the HL-LHC. Signal and background events are generated at the leading order (LO) using MadGraph5_aMC@NLO [57]. During the event generation process, specific generation-level cuts are applied, and their details can be found in Appendix A (refer to Table 8). NNPDF2.3NLO parton distribution function (PDF) set [58] is used. Subsequent showering and hadronization of the generated events are carried out using Pythia8 [59] with the A14 tune [60]. The reconstruction of jets is performed within the FastJet framework [61], utilizing the anti-$kT$ clustering algorithm [62] with a jet parameter $R = 0.4$ and a transverse momentum threshold of $p_T > 15$ GeV. To simulate the detector response, we employ Delphes-3.5.0 [63] with the default HL-LHC ATLAS analysis card.

The primary source of irreducible background contribution in both the b-veto and b-tag category arises from the $W \to \tau\nu$ process (see Fig. 2 (right)). To generate this background, we incorporate two additional jets using the MLM merging scheme [64]. The extra jet can consist of gluon, light quarks, $c$-quark, and $b$-quark. The next dominant background contamination originates from fake-$\tau_h$ events and Drell-Yan (DY) production. We generate the $jj\nu\nu$ process, where the light jets, denoted as $j$, can mimic a hadronic $\tau$ ($\tau_h$). We refer to this background as the "Misid. $\tau_h$" background. To improve upon event statistics, we generate the DY process ($pp \to \tau\tau$) separately in different $m_{\tau\tau}$ mass regions. These regions are subsequently combined with appropriate cross-section factors. For the di-boson (VV) background, we simulate the $WW$, $WZ$, and $ZZ$ processes separately. The DY+jets and VV+jets backgrounds are generated with two extra jets using the MLM merging scheme [64]. A notable contribution to the background arises from top pair ($t\bar{t}$) production due to its large production cross-section. We generate this background in three categories: fully hadronic $t\bar{t}$, semi-leptonic $t\bar{t}$ (where one $W$ boson decays hadronically and the other decays leptonically), and fully leptonic $t\bar{t}$ (where both $W$ bosons decay leptonically). Lastly, we simulate the single-top background by merging it with one additional jet.

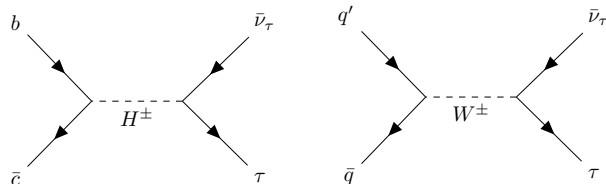

**Figure 2**: *Feynman diagrams for the signal process, $bc \to H^{\pm} \to \tau\nu$ (left) and the dominant irreducible background, $pp \to W^{\pm} \to \tau\nu$ (right), at leading order.*

## 4.1 The $\tau_h \nu$ channel

The objective of this section is to investigate the reach of the HL-LHC in constraining low-mass charged Higgs bosons within the general 2HDM model. We perform an optimized cut-based analysis by varying the charged Higgs mass, considering the following benchmark points: $m_{H^{\pm}} = 180,\ 200,\ 250,\ 300,\ 350$ and $400$ GeV.

Our analysis criteria are as follows: We select events that contain exactly one $\tau$-tagged jet with a transverse momentum ($p_T$) greater than 30 GeV and a pseudorapidity ($|\eta|$) within the range of $|\eta| < 4.0$. Leptons ($e, \mu$) with $p_T$ exceeding 20 GeV and $|\eta|$ within $|\eta| < 4.0$ are vetoed. Events are required to have zero b-tagged jets with $p_T$ greater than 30 GeV and $|\eta|$ within $|\eta| < 4.0$. Furthermore, the total number of light jets in an event should not exceed two. This specific requirement significantly reduces the backgrounds from hadronic and semi-leptonic top pair ($t\bar{t}$) production. After applying the generation level cuts used for signal and background generation (see Appendix A), we proceed with a similar analysis as performed by CMS [55] with regards to the basic selection cuts. We impose a cut on the azimuthal angle separation between the $\tau$-tagged jet and the missing transverse momentum, requiring $\Delta\phi(p_{T,\tau_h}, \not{p}_T) \geq 2.4$. In signal processes, the hadronic $\tau$ and neutrino are produced nearly back-to-back. Consequently, the missing transverse momentum and $p_{T,\tau_h}$ should exhibit small differences, arising from the dilution of neutrino momentum resulting from the $\tau_h$ decay. Therefore, the ratio of $p_{T,\tau_h}$ to $\not{p}_T$ is constrained within the range $0.7 \leq p_{T,\tau_h}/\not{p}_T \leq 1.3$. The normalized kinematic distributions of $\Delta\phi(p_{T,\tau_h}, \not{p}_T)$ and $p_{T,\tau_h}/\not{p}_T$ are presented in Fig. 3 for $m_{H^{\pm}} = 180$ and $400$ GeV, including all relevant backgrounds.

Following the selection cuts, we perform a cut-based analysis by optimizing the cuts on three observables: transverse momentum of the $\tau$-tagged jet ($p_{T,\tau_h}$), transverse mass ($m_T$), and missing transverse energy ($\not{E}_T$). The objective is to maximize the signal significance, defined as $\sigma_s = S/\sqrt{B}$, where $S$ and $B$ represent the signal and background yields at a given integrated luminosity. Table 2 summarizes the trigger, basic selection, and optimized cuts for each observable. It is observed that the optimized cuts become stricter as the charged Higgs mass increases. This is consistent with the trend observed in Fig. 3, where the signal

| Cuts applied | |
| --- | --- |
| Trigger cuts | Basic selection cuts |
| $N_{\tau_h} = 1, \ N_\ell = 0$ <br> b-veto, $N_{\text{b-jets}} = 0$ <br> $N_j \leq 2$ | $\Delta\phi(p_{T,\tau_h}, \not{p}_T) \geq 2.4$ <br> $0.7 \leq p_{T,\tau_h}/\not{p}_T \leq 1.3$ |
| Optimised cuts | |
| $p_{T,\tau_h} \geq [50, 50, 70, 80, 90, 110]$ GeV <br> $m_T \geq [100, 110, 150, 170, 200, 220]$ GeV <br> $\not{E}_T \geq [50, 50, 60, 80, 90, 100]$ GeV | for $m_{H^\pm} = [180, 200, 250, 300, 350, 400]$ GeV |

**Table 2**: *Cuts imposed on the cut-based analysis in $pp \to H^\pm \to \tau_h \nu$ channel.*

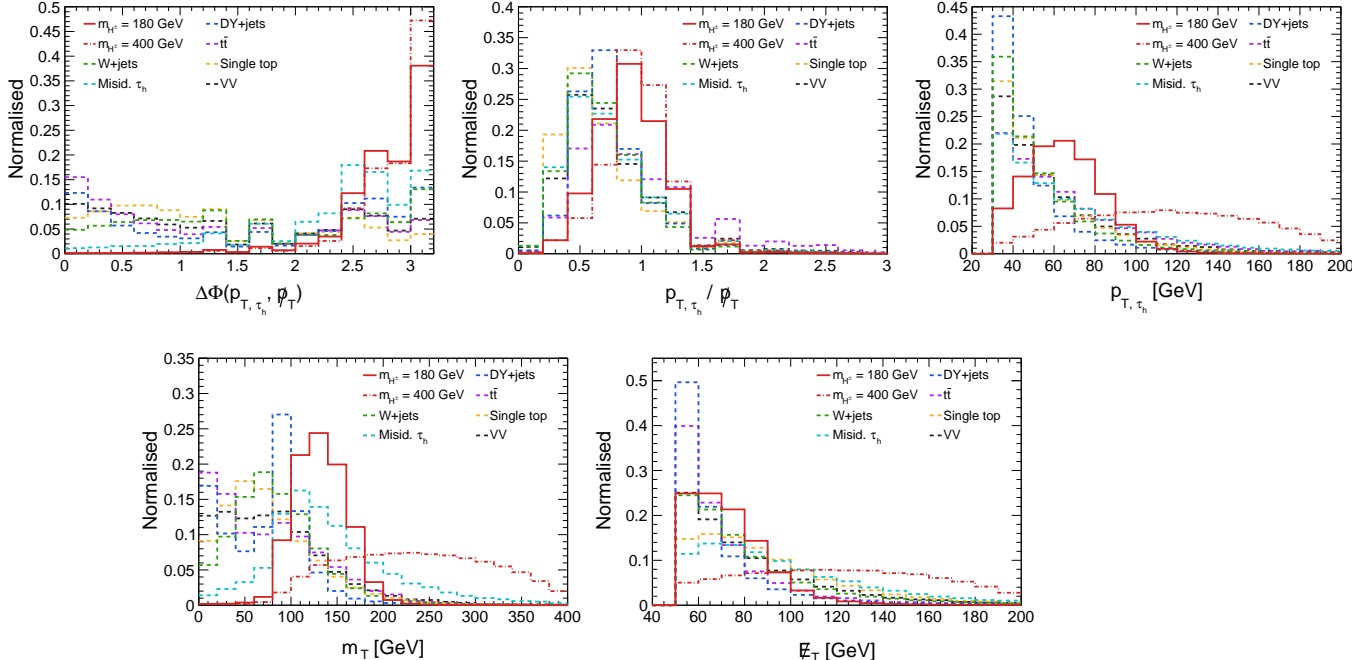

**Figure 3**: *The normalised kinematic distributions of $\Delta\phi(p_{T,\tau_h}, \not{p}_T)$, $p_{T,\tau_h}/\not{p}_T$, $p_{T,\tau_h}$, $m_T$ and $\not{E}_T$ for charged Higgs masses of $m_{H^\pm} = 180$ and $400$ GeV with backgrounds, for the b-veto category. The distributions are shown after the basic trigger and generation level cuts.*

kinematic distributions become flatter with increasing $m_{H^\pm}$, and a stronger cut enhances the separation between the signal and background. To quantify the impact of each cuts, we provide a cut-flow table in Table 3 for $m_{H^\pm} = 180$ and 400 GeV. This table illustrates the number of background events at the HL-LHC and signal efficiency remaining after each stage of the analysis, allowing for a clear understanding of the signal and background contributions at each step.

| Cut flow | Signal Efficiency, $\epsilon$ ($\times 10^{-2}$) | Background rates at $\sqrt{s} = 14$ TeV with $\mathcal{L} = 3$ ab$^{-1}$ | | | | | | | |
|---|---|---|---|---|---|---|---|---|---|
| | | W+jets | Misid. $\tau_h$ | DY+jets | $t\bar{t}$ had | $t\bar{t}$ semi-lep | $t\bar{t}$ lep | VV+jets | Single $t$ |
| | | ($\times 10^7$) | | | ($\times 10^5$) | | | ($\times 10^6$) | |
| $m_{H^\pm} = 180$ GeV | | | | | | | | | |
| Trigger+Gen | 19.9 | 14.9 | 2.0 | 4.3 | 10.1 | 44.5 | 7.8 | 9.7 | 3.1 |
| $N_j \leq 2$ | 19.8 | 13.6 | 1.7 | 3.9 | 2.0 | 22.4 | 6.3 | 7.2 | 2.4 |
| $\Delta\phi(p_{T,\tau_h}, \not{p}_T)$ | 17.8 | 4.7 | 1.1 | 1.7 | 0.6 | 3.2 | 2.2 | 2.1 | 0.5 |
| $p_{T,\tau_h}/\not{p}_T$ | 13.0 | 2.8 | 0.44 | 0.86 | 0.25 | 1.3 | 0.8 | 0.9 | 0.2 |
| $p_{T,\tau_h}$ | 11.5 | 2.2 | 0.41 | 0.43 | 0.20 | 1.1 | 0.75 | 0.77 | 0.16 |
| $m_T$ | 10.7 | 2.1 | 0.41 | 0.40 | 0.19 | 1.1 | 0.72 | 0.75 | 0.16 |
| $\not{E}_T$ | 10.7 | 2.1 | 0.41 | 0.40 | 0.19 | 1.1 | 0.72 | 0.75 | 0.16 |
| $m_{H^\pm} = 400$ GeV | | | | | | | | | |
| Trigger+Gen | 28.0 | 14.9 | 2.0 | 4.3 | 10.1 | 44.5 | 7.8 | 9.7 | 3.1 |
| $N_j \leq 2$ | 27.7 | 13.6 | 1.7 | 3.9 | 2.0 | 22.4 | 6.3 | 7.2 | 2.4 |
| $\Delta\phi(p_{T,\tau_h}, \not{p}_T)$ | 25.5 | 4.7 | 1.1 | 1.7 | 0.6 | 3.2 | 2.2 | 2.1 | 0.5 |
| $p_{T,\tau_h}/\not{p}_T$ | 20.0 | 2.8 | 0.44 | 0.86 | 0.25 | 1.3 | 0.8 | 0.9 | 0.2 |
| $p_{T,\tau_h}$ | 12.4 | 0.14 | 0.12 | 0.02 | 0.01 | 0.2 | 0.14 | 0.12 | 0.03 |
| $m_T$ | 11.0 | 0.12 | 0.11 | 0.01 | 0.008 | 0.18 | 0.12 | 0.11 | 0.02 |
| $\not{E}_T$ | 11.0 | 0.12 | 0.11 | 0.01 | 0.008 | 0.18 | 0.12 | 0.11 | 0.02 |

**Table 3**: *Signal efficiency and background yields at the HL-LHC, in the $pp \to H^\pm \to \tau_h \nu$ channel, at each step of the cut-based analysis for the charged Higgs mass of $m_{H^\pm} = 180$ and 400 GeV.*

As previously mentioned, the $N_j \leq 2$ cut effectively reduces the $t\bar{t}$ background, resulting in a significant suppression. The $\Delta\phi(p_{T,\tau_h}, \not{p}_T)$ cut plays a crucial role in reducing all backgrounds while minimally affecting the signal process. For $m_{H^\pm} = 400$ GeV, the optimized cuts on $p_{T,\tau_h}$, $m_T$, and $\not{E}_T$ lead to a reduction in all backgrounds by approximately one to two orders of magnitude. In Table 4, we provide the background yields and signal efficiencies after the cut-based analysis for $m_{H^\pm} = 200$, 250, 300, and 350 GeV. Indeed, for lower values of $m_{H^\pm} = 180$ GeV, the W+jets background contamination is significant, accounting for approximately 70% of the total background yield. The Misid. $\tau_h$ and DY+jets backgrounds contribute approximately 13% each. However, as $m_{H^\pm}$ increases, the signal and backgrounds become better separated, as evident from the kinematic distributions of $p_{T,\tau_h}$, $m_T$, and $\not{E}_T$ in Fig. 3. This leads to a reduction in the W+jets background yield, and the contribution from Misid.$\tau_h$ becomes closer to the dominant W+jets background. For instance, after the cut-based analysis, the W+jets and Misid. $\tau_h$ backgrounds contribute approximately 47% and 42% to the total background yield, respectively, for $m_{H^\pm} = 400$ GeV.

To calculate the projected upper limit on the charged Higgs production cross-section in a model-independent way, we use the following formula:

| $pp \to H^{\pm} \to \tau_h \nu$, $\sqrt{s} = 14$ TeV | | | | | | | | |
|---|---|---|---|---|---|---|---|---|
| Masses | Background yields after the cut-based analysis at 3 $ab^{-1}$ | | | | | | | Signal Efficiency, |
| $m_{H^{\pm}}$ | W+jets | Misid. $\tau_h$ | DY+jets | $t\bar{t}$ had | $t\bar{t}$ semi-lep | $t\bar{t}$ lep | VV+jets | Single $t$ | $\epsilon$ |
| (GeV) | ($\times 10^7$) | | | ($\times 10^5$) | | | ($\times 10^6$) | | ($\times 10^{-2}$) |
| 200 | 1.9 | 0.4 | 0.3 | 0.2 | 1.1 | 0.7 | 0.7 | 0.15 | 11.6 |
| 250 | 0.6 | 0.3 | 0.06 | 0.05 | 0.6 | 0.4 | 0.35 | 0.08 | $\sim 10$ |
| 300 | 0.36 | 0.2 | 0.04 | 0.03 | 0.4 | 0.3 | 0.25 | 0.06 | 10.9 |
| 350 | 0.2 | 0.16 | 0.02 | 0.02 | 0.3 | 0.2 | 0.16 | 0.04 | 10.9 |

**Table 4**: *Signal efficiency and background yields after the cut-based analysis in the $pp \to H^{\pm} \to \tau_h \nu$ channel at the HL-LHC.*

$$\sigma(pp \to H^{\pm} \to \tau_h \nu)_{\text{UL}} = \frac{N \cdot \sqrt{B}}{\epsilon \cdot \mathcal{L}} \tag{4.1}$$

where:

- $\sigma(pp \to H^{\pm} \to \tau_h \nu)_{\text{UL}}$ is the projected upper limit on the charged Higgs production cross-section in the $\tau_h \nu$ final state.

- $N$ is the number of confidence intervals or the desired significance level. For example, a 95% confidence level corresponds to $N = 2$.

- $B$ is the total background yield, which is the sum of all background events after the cut-based analysis.

- $\epsilon$ is the signal efficiency, representing the fraction of signal events that pass the selection cuts.

- $\mathcal{L}$ is the integrated luminosity, which denotes the total amount of data collected.

By plugging in the appropriate values for $N$, $B$, $\epsilon$, and $\mathcal{L}$ from our analysis, we calculate the projected upper limit on the charged Higgs production cross-section. This provides an estimate of the maximum allowed cross-section for the charged Higgs production in the $\tau_h \nu$ final state based on the analysis results and the chosen confidence level.

We evaluate the $\sigma(pp \to H^{\pm} \to \tau_h \nu)_{\text{UL}}$ at different significance levels, such as $2\sigma$ (exclusion limit) and $5\sigma$ (discovery limit). The derived upper limits are shown as solid green and purple lines in Fig. 4. For example, for the $2\sigma$ upper limit, the values vary in the range [32.26 : 7.38] fb for $m_{H^{\pm}} = [180 : 400]$ GeV. These values represent the maximum allowed cross-section for the charged Higgs production in the $\tau_h \nu$ final state at a 95% confidence level. In addition, a systematic uncertainty of 2% is included in the analysis. The resulting upper

limits, accounting for this systematic uncertainty, are shown as dashed colored lines in Fig. 4. It is worth noting that the presence of systematics can weaken the limits due to the significant contamination from the W+jets background, which constitutes a substantial fraction of the total background (approximately $50-70\%$) in this channel.

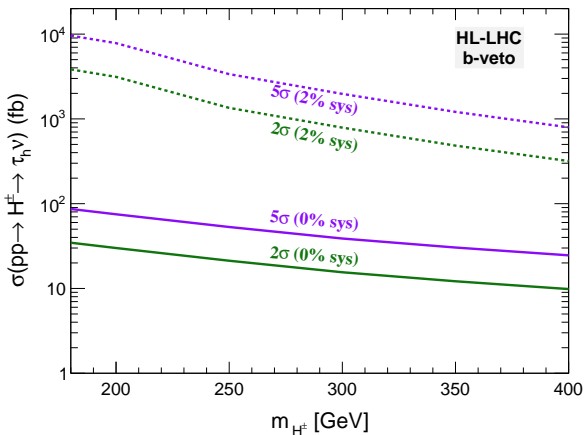

**Figure 4**: *Upper limit on $\sigma(pp \to H^\pm \to \tau\nu)$ as a function of $m_{H^\pm}$ in the b-veto category. The green and purple solid (dashed) lines show the $2\sigma$ and $5\sigma$ upper limit upon including $0\%$ (2%) systematic uncertainties.*

## 4.2 The $b\tau_h\nu$ channel

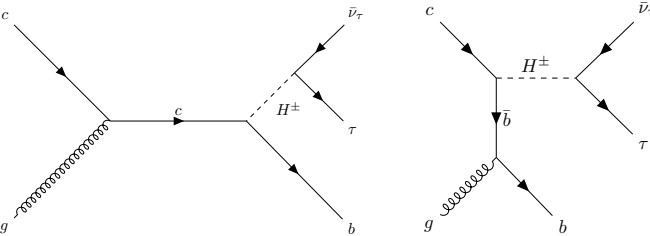

**Figure 5**: *The Feynman diagrams for the signal process $gc \to bH^\pm \to b\tau\nu$ in the b-tag category at leading order.*

In the $pp \to bH^\pm \to b\tau\nu$ channel, the charged Higgs boson is produced in association with a bottom quark. This final state is characterized by the presence of one b-tagged jet, along with the hadronically decaying tau lepton ($\tau_h$) and missing transverse energy ($\not{E}_T$). The leading order Feynman diagrams for the signal process are shown in Fig. 5. We select events with exactly one $\tau$-tagged jet with $p_T > 30$ GeV and $|\eta| < 4.0$. Leptons are vetoed having $p_T > 20$ GeV and $|\eta| < 4.0$. Events are required to have exactly one b-tagged jet with

$p_T > 30$ GeV and $|\eta| < 4.0$. We further restrict the total number of light jets to be at most two.

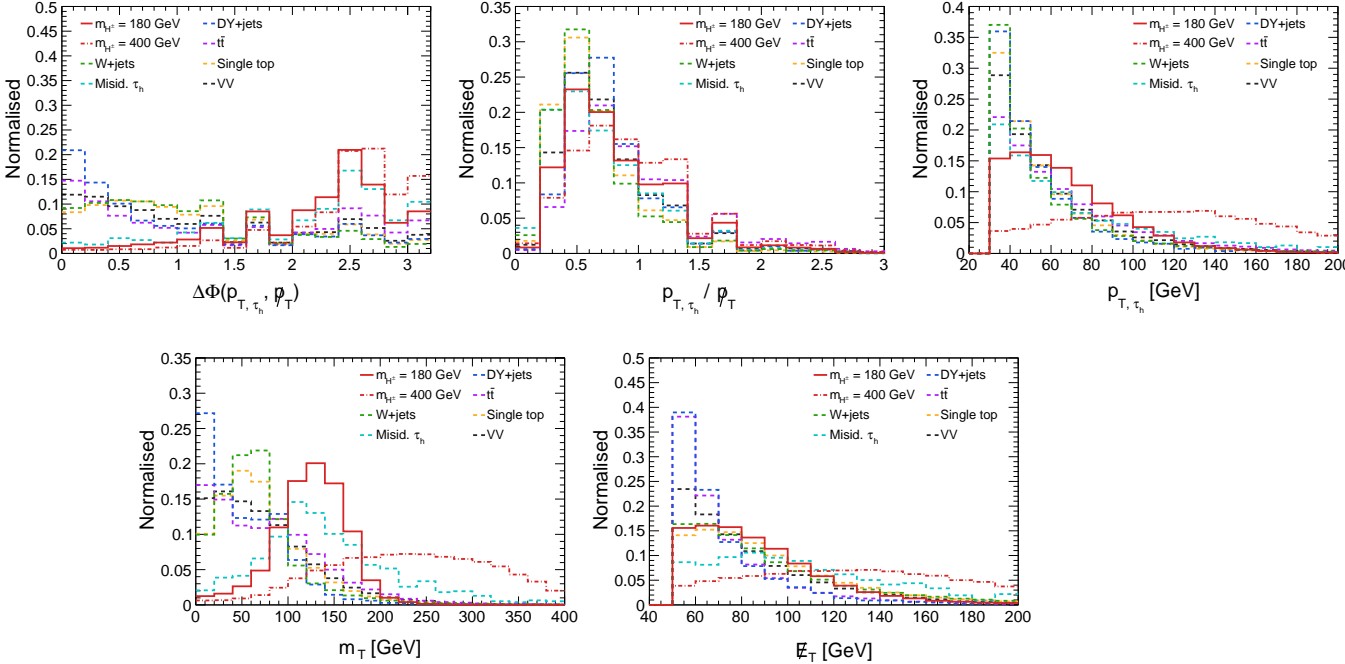

**Figure 6**: *The normalised kinematic distributions of $\Delta\phi(p_{T,\tau_h}, \not{p}_T)$, $p_{T,\tau_h}/\not{p}_T$, $p_{T,\tau_h}$, $m_T$ and $\not{E}_T$ for charged Higgs masses of $m_{H^\pm} = 180$ and $400$ GeV with backgrounds, for the b-tag category. The distributions are shown after the basic trigger and generation level cuts.*

The normalised kinematic distributions of $\Delta\phi(p_{T,\tau_h}, \not{p}_T)$, $p_{T,\tau_h}/\not{p}_T$, $p_{T,\tau_h}$, $m_T$, and $\not{E}_T$ for the $pp \to bH^\pm \to b\tau\nu$ channel are shown in Fig. 6. From the distributions, we observe that the signal and backgrounds overlap for the $p_{T,\tau_h}/\not{p}_T$ distribution. This can be understood by considering the additional b-jet present in the signal process. The charged Higgs recoils against the b-jet, which further decays to $\tau\nu$. The momentum imbalance of the b-jet and the hadronically decaying $\tau$ lepton contributes to the total missing transverse energy ($\not{E}_T$) in the event. As a result, the overall $\not{E}_T$ is higher compared to the previous b-veto signal category. The transverse momentum of the $\tau$ lepton is slightly lower in this final state. These factors lead to a decrease in the ratio $p_{T,\tau_h}/\not{p}_T$, causing the signal distribution to overlap with the background processes. Therefore, we exclude this observable from further optimization. Instead, we focus on four kinematic observables for the cut-based optimization: $\Delta\phi(p_{T,\tau_h}, \not{p}_T)$, $p_{T,\tau_h}$, $m_T$, and $\not{E}_T$. These observables exhibit better discrimination between the signal and background processes. The trigger-level cuts and the optimized cuts for these observables are summarized in Table 5. These cuts are chosen to enhance the signal significance and improve

the sensitivity of the analysis.

| Cuts applied | |
|---|---|
| Trigger cuts | |
| $N_{\tau_h} = 1, \ N_\ell = 0$ | |
| b-tag, $N_{\mathrm{b-jets}} = 1$ | |
| $N_j \leq 2$ | |
| Optimised cuts | |
| $p_{T,\tau_h} \geq [30, 30, 40, 50, 50, 60]$ GeV $\Delta\phi(p_{T,\tau_h}, \not{p}_T) \geq [1.1, 1.0, 0.9, 0.7, 1.0, 1.0]$ $m_T \geq [100, 110, 140, 170, 200, 230]$ GeV $\not{E}_T \geq [50, 50, 60, 70, 80, 100]$ GeV | for $m_{H^\pm} = [180, 200, 250, 300, 350, 400]$ GeV |

**Table 5**: *Trigger-level and optimised cuts imposed on the cut-based analysis in b-tag category.*

The signal efficiency and background yields after each optimized cut for the $pp \rightarrow bH^\pm \rightarrow b\tau\nu$ channel at an integrated luminosity of 3 ab$^{-1}$ are presented in Table 6. The most effective observable in reducing the backgrounds is the transverse mass, $m_T$. The cut on $m_T$ becomes stronger as the charged Higgs mass increases, resulting in a reduction of the dominant W+jets background by approximately one order of magnitude. The $\Delta\phi(p_{T,\tau_h}, \not{p}_T)$ variable is also effective in suppressing all background processes with a negligible impact on the signal efficiency. It helps to enhance the signal-to-background ratio by exploiting the back-to-back nature of the signal process, where the hadronically decaying $\tau$ and the missing transverse momentum are almost in opposite directions with dilution coming from the additional b-jet in this b-tag category. In Table 7, we provide the signal efficiency and background yields at a center-of-mass energy of $\sqrt{s} = 14$ TeV for charged Higgs mass, $m_{H^\pm} = 200$, 250, 300, and 350 GeV. These results demonstrate the effectiveness of the optimized cuts in enhancing the signal significance and reducing the background contributions.

Finally, we investigate the upper limits on the production cross-section, denoted as $\sigma(pp \rightarrow bH^\pm \rightarrow b\tau_h\nu)_{UL}$, in relation to the charged Higgs mass. The corresponding results are illustrated in Fig. 7. For the charged Higgs mass range of 180 GeV $\leq m_{H^\pm} \leq$ 400 GeV, the 95% confidence level (CL) upper limit, without any systematic uncertainties, exhibits a variation from 31.89 fb to 6.84 fb. When considering a 2% systematic uncertainty, the $2\sigma$ upper limit is approximately 70.26 fb for $m_{H^\pm} = 400$ GeV. Despite the additional b-tagging requirement favoring the $t\bar{t}$ background, this condition significantly suppresses the dominant W+jets background. Consequently, the upper limit in this channel becomes more stringent compared to the previously discussed $pp \rightarrow H^\pm \rightarrow \tau\nu$ final state due to a higher signal-to-background ratio (S/B).

| Cut flow | Signal Efficiency, $\epsilon$ ($\times 10^{-2}$) | Background rates at $\sqrt{s}=14$ TeV with $\mathcal{L}=3$ ab$^{-1}$ | | | | | | | |
|---|---|---|---|---|---|---|---|---|---|
| | | W+jets | Misid. $\tau_h$ | DY+jets | $t\bar{t}$ had | $t\bar{t}$ semi-lep | $t\bar{t}$ lep | VV+jets | Single $t$ |
| | | ($\times 10^6$) | | | ($\times 10^5$) | | | ($\times 10^6$) | |
| $m_{H^\pm}=180$ GeV | | | | | | | | | |
| Trigger+Gen | 7.0 | 4.8 | 0.6 | 1.8 | 18.3 | 83.5 | 15.6 | 8.0 | 5.2 |
| $N_j \leq 2$ | 6.9 | 4.4 | 0.5 | 1.6 | 5.3 | 52.9 | 13.7 | 7.2 | 4.6 |
| $p_{T,\tau_h}$ | 6.9 | 4.4 | 0.5 | 1.6 | 5.3 | 52.9 | 13.7 | 7.2 | 4.6 |
| $\Delta\phi(p_{T,\tau_h},\not{p}_T)$ | 6.3 | 2.0 | 0.44 | 0.6 | 3.0 | 21.5 | 9.7 | 3.5 | 2.2 |
| $m_T$ | 4.8 | 0.6 | 0.36 | 0.2 | 1.8 | 9.4 | 6.0 | 1.8 | 1.0 |
| $\not{E}_T$ | 4.8 | 0.6 | 0.36 | 0.2 | 1.8 | 9.4 | 6.0 | 1.8 | 1.0 |
| $m_{H^\pm}=400$ GeV | | | | | | | | | |
| Trigger+Gen | 13.0 | 4.8 | 0.6 | 1.8 | 18.3 | 83.5 | 15.6 | 8.0 | 5.2 |
| $N_j \leq 2$ | 12.3 | 4.4 | 0.5 | 1.6 | 5.3 | 52.9 | 13.7 | 7.2 | 4.6 |
| $p_{T,\tau_h}$ | 10.8 | 1.3 | 0.3 | 0.5 | 2.7 | 18.0 | 4.5 | 2.6 | 1.5 |
| $\Delta\phi(p_{T,\tau_h},\not{p}_T)$ | 10.4 | 0.5 | 0.2 | 0.2 | 1.6 | 7.6 | 3.5 | 1.3 | 0.7 |
| $m_T$ | 5.8 | 0.06 | 0.07 | 0.006 | 0.2 | 0.7 | 0.43 | 0.1 | 0.06 |
| $\not{E}_T$ | 5.5 | 0.05 | 0.07 | 0.003 | 0.07 | 0.6 | 0.4 | 0.1 | 0.06 |

**Table 6**: *Signal efficiency and background yields at the HL-LHC, in the $pp \to bH^\pm \to b\tau_h\nu$ channel, at each step of the cut-based analysis for the charged Higgs mass of $m_{H^\pm}=180$ and $400$ GeV.*

| | $pp \to bH^\pm \to b\tau_h\nu$, $\sqrt{s}=14$ TeV | | | | | | | | |
|---|---|---|---|---|---|---|---|---|---|
| Masses | Background yields after the cut-based analysis at 3 $ab^{-1}$ | | | | | | | | Signal Efficiency, |
| $m_{H^\pm}$ | W+jets | Misid. $\tau_h$ | DY+jets | $t\bar{t}$ had | $t\bar{t}$ semi-lep | $t\bar{t}$ lep | VV+jets | Single $t$ | $\epsilon$ |
| (GeV) | ($\times 10^6$) | | | ($\times 10^5$) | | | ($\times 10^6$) | | ($\times 10^{-2}$) |
| 200 | 0.5 | 0.3 | 0.15 | 1.5 | 7.7 | 5.0 | 1.5 | 0.8 | 5.3 |
| 250 | 0.3 | 0.2 | 0.05 | 0.6 | 3.9 | 2.5 | 0.7 | 0.4 | 5.4 |
| 300 | 0.14 | 0.15 | 0.02 | 0.3 | 2.0 | 1.3 | 0.4 | 0.2 | 5.5 |
| 350 | 0.08 | 0.1 | 0.009 | 0.16 | 1.1 | 0.7 | 0.2 | 0.1 | 5.6 |

**Table 7**: *Signal efficiency and background yields after the cut-based analysis in the $pp \to bH^\pm \to b\tau_h\nu$ channel at the HL-LHC.*

# 5 Collider prospect of flavor anomalies

In the present section, we analyze the ramifications arising from the implementation of upper limits on the cross-section of charged Higgs production, focusing on their impact on the Yukawa parameter space as well as the extent to which the HL-LHC can effectively probe the diverse anomalies observed in the $b \to c\tau\nu$ decay processes.

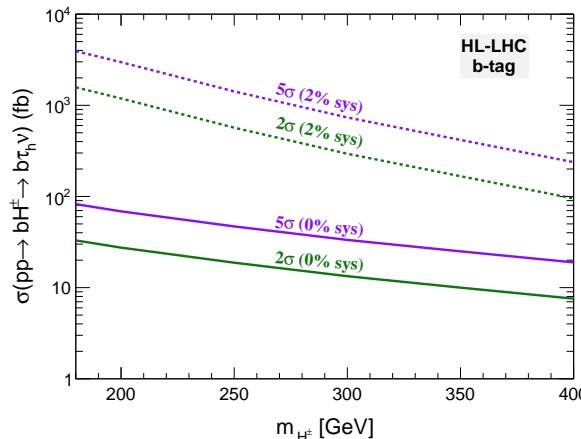

**Figure 7**: *Upper limit on $\sigma(pp \to H^{\pm} \to \tau\nu)$ as a function of charged Higgs mass in the b-tag category. The green and purple solid (dashed) lines show the $2\sigma$ and $5\sigma$ upper limit upon including $0\%$ ($2\%$) systematic uncertainties.*

## 5.1 The Yukawa coupling plane

From Eq. 3.3, it is evident that the scalar coupling $C_{RR}^{S}$ is directly proportional to the product of the charged Higgs Yukawa couplings, $\tilde{y}_{bc} \times \tilde{y}_{\tau\nu}$. The objective of this section is to scrutinize the two-dimensional parameter space of Yukawa couplings, namely $|\tilde{y}_{bc}| - |\tilde{y}_{\tau\nu}|$. For our analysis, we consider the charged Higgs mass to lie within the range of [180, 400] GeV. In Fig. 8, we show the $1\sigma$ and $2\sigma$ allowed region of Yukawa couplings $|\tilde{y}_{bc}|$ and $|\tilde{y}_{\tau\nu}|$ favored by $R_D$, $R_{D^{\star}}$, $P_{\tau}^{D^{\star}}$, $F_{L}^{D^{\star}}$, $R_{J/\psi}$, and $R_{\Lambda_c}$, represented by the darker and light yellow band, respectively. These allowed regions are obtained by minimizing $\chi^2$ of a negative log-likelihood function. The bands demonstrate consistency of the Yukawa coupling product, $\tilde{y}_{bc} \times \tilde{y}_{\tau\nu}$, with current measurements of all the considered flavor observables. It is evident that the $|\tilde{y}_{bc}| - |\tilde{y}_{\tau\nu}|$ parameter space is more constrained for low mass charged Higgs boson. Additionally, the region below the solid blue line corresponds to the allowed parameter space considering a $\mathcal{B}(B_c \to \tau\nu)$ branching ratio of 63%. Further, the large $\tilde{y}_{bc}$ values are subject to constraints from $B_s - \bar{B}_s$ mixing, as illustrated by the excluded gray colored region in the figure. We account for the renormalization group equation (RGE) running effect of the strong coupling constant $\alpha_s$ [65]. The contribution of the charged Higgs to the $B_s - \bar{B}_s$ mixing, occurring at the 1-loop level, is taken from [26], and we follow [66] to impose constraint on this parameter. The constraint is stronger (larger gray colored region) for low mass charged Higgs boson and it reduces the allowed $|\tilde{y}_{bc}| - |\tilde{y}_{\tau\nu}|$ parameter space significantly. Finally, we incorporate the 95% confidence level (CL) upper limits obtained from the $\tau_h\nu$ (b-veto category) and $b\tau_h\nu$ (b-tag category) searches discussed in the previous section. These upper

limits are represented by the green and purple colored lines, respectively, in Fig. 8. The collider search, specifically b-tag analysis can probe the remaining region of parameter space in $|\tilde{y}_{bc}| - |\tilde{y}_{\tau\nu}|$ plane for higher charged Higgs masses.

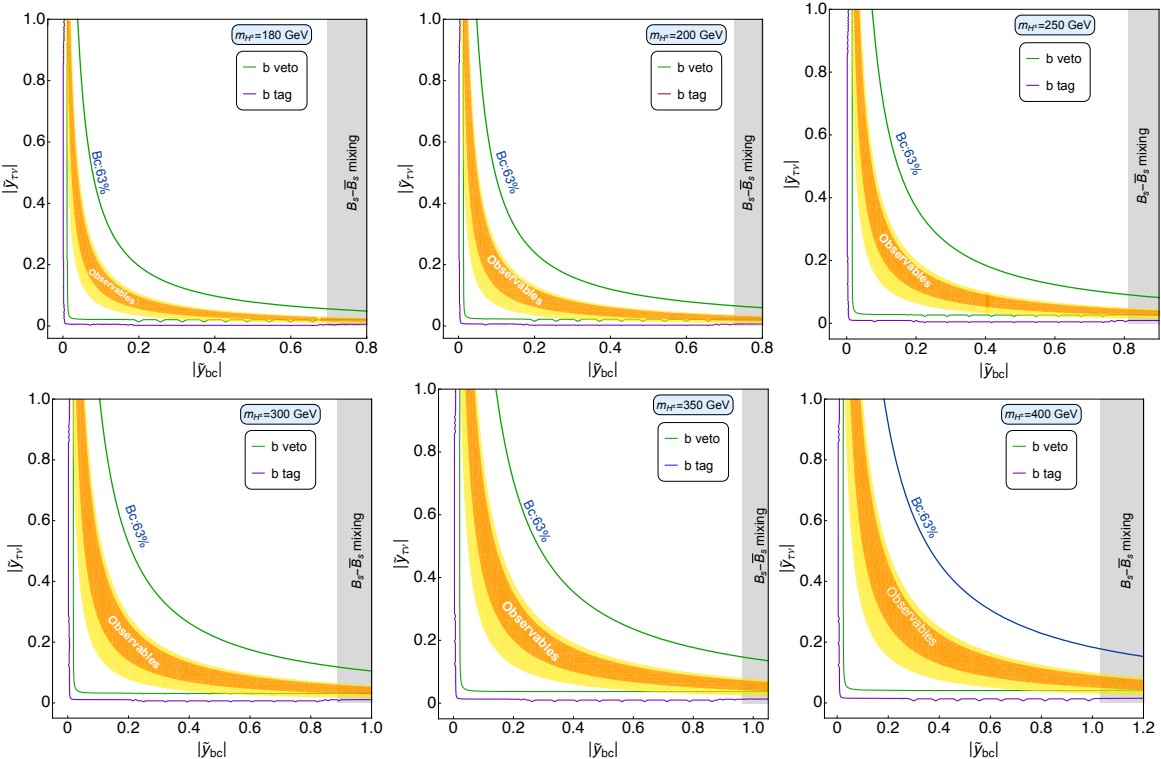

**Figure 8**: *Constraints on $|\tilde{y}_{bc}| - |\tilde{y}_{\tau\nu}|$ parameter space for various charged Higgs masses. The darker yellow and light yellow bands indicate the $1\sigma$ and $2\sigma$ region in consonant with the current measurement of corresponding flavor observables. Here, Bc represents the $\mathcal{B}(B_c \to \tau\nu)$. The collider constraints are shown with green and purple lines for b-veto and b-tag analysis, respectively.*

## 5.2   Accessible region in the flavor observable planes

In Figure 9, we illustrate the flavor observables in several 2D planes, similar to Fig. 1. The black star denotes the SM prediction. The experimental central value is represented by the blue star and the corresponding $1\sigma$ and $2\sigma$ uncertainty is represented with solid and dashed blue contour, respectively. The best fit value of $C_{RR}^S$ coupling is denoted with a red circle. Next, we evaluate the accessible region in these flavor observable planes in case of charged Higgs coupled to a right-handed neutrino in G2HDM model and it is shown with the solid green line. The points on the green line are the only accessible values within the current model, which passes through the SM and the best fit value of $C_{RR}^S$ coupling. The line is

evaluated by solving for Yukawa couplings after inserting $C_{RR}^S$ of Eq. 3.3 into Eq. 2.2. For large $\tilde{y}_{bc}$ values, the constraint from b-veto category almost falls within the $2\sigma$ contour of flavor observables (see Fig. 8). Therefore, we select two benchmark points for b-tag category based on the collider constraints obtained in the $|\tilde{y}_{bc}| - |\tilde{y}_{\tau\nu}|$ plane. Specifically, we consider the points $(|\tilde{y}_{bc}|, |\tilde{y}_{\tau\nu}|) \sim (0.6, 0.022)$ and $(0.9, 0.016)$ for $m_{H^\pm} = 180$ and $400$ GeV, respectively. These constraints are then translated into the 2D plane of flavor observables, as shown in Figure 9. In the figure, we represent the benchmark points for the 180 and 400 GeV with purple and yellow-colored tick symbol, respectively. A significant part of the green line and the HL-LHC accessible points lie within the $2\sigma$ contour. To summarize, current measurement of flavor observables is within the reach of the considered model at $2\sigma$ uncertainty.

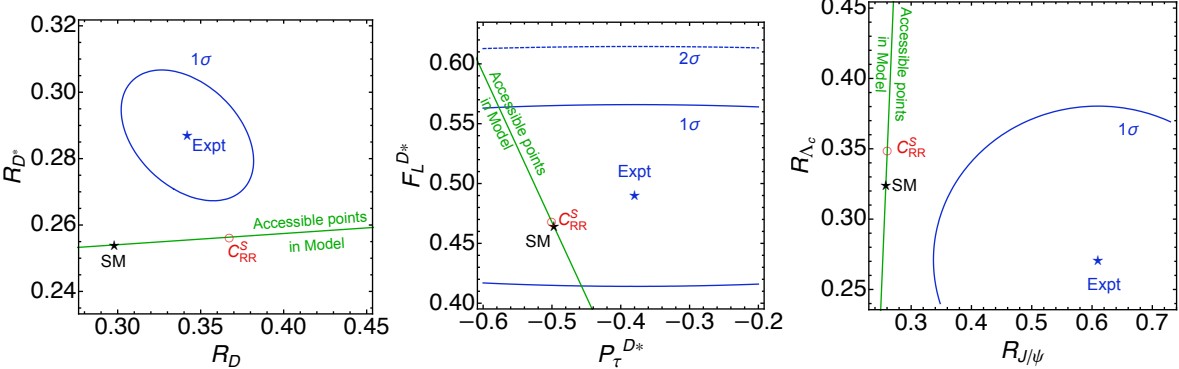

**Figure 9**: *Accessible region in the flavor observable planes. The points on the green colored line are accessible in the present model.*

## 6  Summary and outlook

Experimental measurements of various flavor ratios, such as $R_D$, $R_{D^\star}$, $P_\tau^{D^\star}$, $F_L^{D^\star}$, $R_{J/\psi}$, and $R_{\Lambda_c}$ exhibit deviations from SM. If these deviations persist in future experiments, it would provide unequivocal evidence of physics beyond the SM.

In this study, we explore these anomalies by considering an effective Lagrangian that incorporates new physics coming from right handed neutrinos. In order to determine the NP coupling, we perform a $\chi^2$ fit by including six measurements, namely $R_D$, $R_{D^\star}$, $P_\tau^{D^\star}$, $F_L^{D^\star}$, $R_{J/\psi}$ and $R_{\Lambda_c}$. We obtain the best fit value to be $C_{RR}^S/C_{LR}^S = -0.51$. We analyzed the impact of the NP scalar couplings on $R_D - R_{D^\star}$, $P_\tau^{D^\star} - F_L^{D^\star}$ and $R_{J/\psi} - R_{\Lambda_c}$. From the $R_D$-$R_{D^\star}$ plane, we observe that although the best-fit value lies within the $2\sigma$ contour, the $1\sigma$ region around the best fit value overlaps with the $1\sigma$ experimental contour. Further, we observe that the range of $\mathcal{B}(B_c \to \tau\nu)$ obtained with $C_{RR}^S/C_{LR}^S$ NP coupling satisfy the $\mathcal{B}(B_c \to \tau\nu) \leq 63\%$ constraint.

We further study the flavor anomalies in case of charged Higgs coupled to a right-handed neutrino in the context of G2HDM model. We perform a collider study with charged Higgs boson in order to explain the $b \to c\tau\nu$ anomalies in presence of right-handed scalar coupling, in the following two final states: $pp \to H^{\pm} \to \tau\nu$ and $pp \to bH^{\pm} \to b\tau\nu$, at a center of mass energy of $\sqrt{s} = 14$ TeV, with an integrated luminosity of 3 $ab^{-1}$. Our study involves a comprehensive collider analysis, wherein we optimize the selection cuts on various kinematic observables for a set of benchmark masses for the charged Higgs boson, specifically $m_{H^{\pm}} = [180, 200, 250, 300, 350, 400]$ GeV. The $pp \to bH^{\pm} \to b\tau\nu$ final state, also referred to as the b-tag category, exhibits lower contamination from the dominant $W$+jets background. Consequently, the addition of systematic uncertainties has a lesser impact on this channel. This advantage can be attributed to the additional requirement of b-tagging in the $b\tau\nu$ final state, which aids in reducing background contributions. Next, we show that the allowed Yukawa plane, $|\tilde{y}_{bc}| - |\tilde{y}_{\tau\nu}|$, is heavily constrained with recent experimental measurements. The upper limits obtained for $\sigma(pp \to bH^{\pm} \to b\tau\nu)$ impose a more stringent constraint on the Yukawa plane compared to the upper limits on $\sigma(pp \to H^{\pm} \to \tau\nu)$ from b-veto category. Additionally, we explore the accessible regions for the model in these flavor observable planes. From our analysis, we observe that the accessible values of the flavor observables in G2HDM model lie within the $2\sigma$ region of the experimental measurement. Further, HL-LHC has the potential to exclude these currently allowed regions.

## Acknowledgments

We would like to thank N. Rajeev, Girish Kumar, Syuhei Iguro, Biplob Bhattacherjee and Ayan Paul for helpful discussions throughout the course of this work. We would also like to thank Wolfgang Altmannshofer for the fruitful discussion on the topics addressed in this article. A.A. thanks the Planck conference, University of Warsaw for hospitality where discussions and part of this work was completed. AA received support from the French government under the France 2030 investment plan, as part of the Initiative d'Excellence d'Aix-Marseille Université - A*MIDEX.

## A Detail of the generation cuts and production cross section of backgrounds

| Process | Backgrounds | Generation-level cuts ($\ell = e^\pm, \mu^\pm, \tau^\pm$) (NA : Not Applied) | Cross section (fb) |
|---|---|---|---|
| $\tau_h \nu$ / $b\tau_h \nu$ | $t\bar{t}$ hadronic[1] | $p_{T,j/b} > 20$ GeV, $p_{T,\ell} > 15$ GeV, $\|\eta_{j/b/\ell}\| < 5.0$, $\Delta R_{j,b,\ell} > 0.2$, $\not{E}_T > 50$ GeV | 199840.6 |
| | $t\bar{t}$ semi-leptonic | same as $t\bar{t}$ hadronic | 41051.6 |
| | $t\bar{t}$ leptonic | | 5483.4 |
| | W + jets | | 672182 |
| | Misid. $\tau_h$ | | 497287 |
| | DY + jets | | 1584131.4 |
| | ZZ + jets | | 12276.8 |
| | WZ + jets | | 41286.5 |
| | WW + jets | | 360887 |
| | Single top Wt-channel | | 26489.8 |

**Table 8**: *The generation level cuts for various backgrounds used in the analyses along with the production cross-sections.*

## B Detail of the input parameters

| Parameters | Values | Parameters | Values |
|---|---|---|---|
| $m_{B^-}$ | 5.27931 | $m_{B_c}$ | 6.2751 |
| $m_{J/\Psi}$ | 3.0969 | $m_{D^{*0}}$ | 2.00685 |
| $m_{\Lambda_b}$ | 5.61951 | $m_{\Lambda_c}$ | 2.28646 |
| $\tau_{B_c}$ | $0.507 \times 10^{-12}$ | $f_{B_c}$ | 0.434(0.015) |
| $\tau_{B^-}$ | $1.638 \times 10^{-12}$ | $m_{D^{*0}}$ | 2.00685 |
| $\tau_{\Lambda_b}$ | $(1.466 \pm 0.010) \times 10^{-12}$ | | |
| $m_{B_c}$ | 6.272 | $m_{B_c^\star}$ | 6.332 |
| $m_e$ | $0.5109989461 \times 10^{-3}$ | $m_\tau$ | 1.77682 |
| $m_b$ | 4.18 | $m_c$ | 0.91 |
| $V_{cb}$ | 0.0409(11) | $G_F$ | $1.1663787 \times 10^{-5}$ |

**Table 9**: *Input parameters [68]. Masses are quoted in units of GeV and lifetimes are shown in units of second, at the renormalization scale $\mu = m_b$.*

---

[1]The $t\bar{t}$ production cross-sections are taken at NNLO order [67].

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
