# Peer review of "Status of G2HDM with right handed neutrino coupling in the light of $b\to c \tau\nu$ anomalies"

_SciPost Physics_

## Round 1 · Referee Report · Anonymous (Referee 1) · 2025-5-8

Strengths

Few, if any. Please refer to the referee's report for details.

Weaknesses

The primary weakness of the manuscript is its limited scientific contribution and lack of originality. For further details, please refer to the report.

Report

The manuscript scipost_202503_00021v1 presents an analysis of the $b \to c$ flavor anomalies within the framework of a generic Two Higgs Doublet Model extended by right-handed neutrinos. The study explores the potential of the High-Luminosity Large Hadron Collider (HL-LHC) to probe the model's parameter space through searches for $\tau \nu$ and $\tau \nu b$ final states. Below, I provide my assessment, highlighting key comments and questions that address the manuscript's shortcomings:

1) In the abstract, the authors write ``The LHCb collaboration has reported a significant deviation, exceeding $3.2 \sigma$, [$\ldots$]''. Where does this number come from? In the LHCb combination

@article{LHCb:2023zxo, author = "Aaij, Roel and others", collaboration = "LHCb", title = "{Measurement of the ratios of branching fractions $\mathcal{R}(D^{*})$ and $\mathcal{R}(D^{0})$}", eprint = "2302.02886", archivePrefix = "arXiv", primaryClass = "hep-ex", reportNumber = "LHCb-PAPER-2022-039, CERN-EP-2022-284", doi = "10.1103/PhysRevLett.131.111802", journal = "Phys. Rev. Lett.", volume = "131", pages = "111802", year = "2023" }

I find a discrepancy of $1.9\sigma$. Isn’t the $3.2\sigma$ value actually the HFLAV average? For reference, see for example: \href{https://hflav-eos.web.cern.ch/hflav-eos/semi/moriond24/html/RDsDsstar/RDRDs.html}{https://hflav-eos.web.cern.ch/ hflav-eos/semi/moriond24/html/RDsDsstar/RDRDs.html}.

A related point is that the text --- see, for example, page 1 --- initially quotes a $3.2\sigma$ discrepancy, which later appears as $3.31\sigma$. In fact, $3.31\sigma$ reflects the current HFLAV average. I suggest the authors clarify and update this for consistency and accuracy.

2) Ref. [9] should be updated, as the corresponding work has already appeared as arXiv:2302.02886 and has been published in PRL. See the related comment above.

3) A minor formatting point: in the last two expressions of Eq. (2.2), the Wilson coefficients $C_{RR}^S$ and $C_{LR}^S$ are written as ${\rm C_{RR}^S}$ and ${\rm C_{LR}^S}$, whereas the standard notation $C_{RR}^S$ and $C_{LR}^S$ is used elsewhere. Consistency would be preferable.

4) In the first unnumbered equation on page 5 of the manuscript, the expression $C_{RR}^S/C_{LR}^S = -0.48 \pm 0.08$'' is intended to indicate that the best-fit values of both Wilson coefficients are equal. However, this notation is potentially misleading, as the left-hand side can easily be misinterpreted as a ratio. I suggest changing the notation to avoid confusion. The same clarification should be applied wherever the authors refer to$C_{RR}^S / C_{LR}^S$'', such as in the conclusions on page 18.

5) In Eq. (3.1), the authors introduce the effective interactions that form the basis of their collider study. They initially state that they work within ``[$\ldots$] the framework of a generic Two Higgs Doublet Model (G2HDM), where both Higgs doublets interact with up-type and down-type quarks'', yet they proceed to focus exclusively on the up-type Yukawa couplings, offering only brief and somewhat unclear explanations. I therefore recommend expanding and clarifying the discussion of the effective Hamiltonian. A concise and transparent treatment can be found, for instance, around Eq. (2.5) of Ref. [35].

6) The LHC analysis presented in Section 6 shows substantial overlap with existing studies, particularly Refs. [35, 36]. In my view, the authors do not clearly explain what is novel or original in their approach compared to, for instance, Ref. [36]. Clarifying this point is particularly important given the authors’ stated aim --- supporting their SciPost submission --- to ``provide a novel and synergetic link between different research areas'', as required to meet the journal’s criteria. To be candid, I find it difficult to regard scipost_202503_00021v1 as presenting genuinely novel research.

7) In their collider study, the authors adopt benchmark values of $2\%$ and $0\%$ for the systematic uncertainties. Figures 4 and 7 illustrate that this assumption significantly affects the derived upper limits on the production cross sections for $pp \to H^\pm \to \tau \nu$ and $pp \to b H^\pm \to b \tau \nu$, respectively --- in the caption of Figure 7 it should presumably say ``$\sigma(pp \to b H^\pm \to b \tau \nu)$” rather than “$\sigma(pp \to H^\pm \to \tau \nu)$''. What is the rationale behind the choice of systematic uncertainty values? I raise this point because in Ref. [55], which the authors use as a blueprint for their own analysis, the CMS collaboration reports significantly larger systematic uncertainties. In Ref. [55], the dominant systematic uncertainty impacting the analysis stems from the modelling of the transverse mass ($m_T$) distribution, specifically due to the parton distribution functions used in simulating background processes. This uncertainty increases with $m_T$ and reaches up to $50\%$ in the highest bin ($m_T > 1.8 \,{\rm TeV}$) of the $m_T$ distribution as reported by CMS. Another source of systematic uncertainty arises from corrections to the measured $\tau$ lepton energy scale, which range between $1.5\%$ and $4\%$. Additional sources of uncertainty are also present. I am not suggesting that the systematic uncertainties in the present analysis are as large as $50\%$ --- clearly they are not, as the sensitivity in scipost_202503_00021v1 does not primarily arise from the very high-$m_T$ region. However, I believe that assuming $2\%$ systematic uncertainties is likely too optimistic.

In this context, I would like to note that all background processes in scipost_202503_00021v1 are simulated at leading order (LO) using the MLM merging scheme. For the dominant $W^\pm \to \tau \nu$ background, the matching includes up to two additional jets. Such background predictions are unlikely to carry systematic uncertainties as small as $2\%$. Achieving that level of precision would certainly require reweighting the LO differential cross sections to include next-to-next-to-leading order (NNLO) QCD corrections, and potentially also next-to-leading order electroweak effects. The NNLO QCD corrections have been known for quite some time

@article{Anastasiou:2003ds, author = "Anastasiou, Charalampos and Dixon, Lance J. and Melnikov, Kirill and Petriello, Frank", title = "{High precision QCD at hadron colliders: Electroweak gauge boson rapidity distributions at NNLO}", eprint = "hep-ph/0312266", archivePrefix = "arXiv", reportNumber = "SLAC-PUB-10288, UH-511-1042-03", doi = "10.1103/PhysRevD.69.094008", journal = "Phys. Rev. D", volume = "69", pages = "094008", year = "2004" }

@article{Gavin:2012sy, author = "Gavin, Ryan and Li, Ye and Petriello, Frank and Quackenbush, Seth", title = "{W Physics at the LHC with FEWZ 2.1}", eprint = "1201.5896", archivePrefix = "arXiv", primaryClass = "hep-ph", reportNumber = "ANL-HEP-PR-11-83, NUHEP-TH-12-01, PSI-PR-12-01", doi = "10.1016/j.cpc.2012.09.005", journal = "Comput. Phys. Commun.", volume = "184", pages = "208--214", year = "2013" }

and it should be relatively straightforward to incorporate them into the analysis presented in scipost_202503_00021v1.

8) In Section 5, the authors present HL-LHC constraints on the magnitudes of the two relevant Yukawa couplings. Could the authors clarify which systematic uncertainties were assumed in deriving these bounds? Again there seems to be a non-negligible overlap with results already obtained earlier in the literature. For instance, in Ref. [36] one can find plots that are very similar to those shown in the work at hand. However, given the use of linear axes, it is quite difficult to compare the results effectively. Would it not be more appropriate to use logarithmic axes instead?

In conclusion, given the manuscript's limited scientific contribution and lack of originality, I regret that I cannot recommend it for publication in SciPost. Moreover, issues with presentation and language throughout the manuscript at times impede readability and clarity. This constitutes my final assessment. I would add that, following substantial improvements, scipost_202503_00021v1 may become suitable for publication in a journal with less stringent scientific criteria.

Requested changes

See report for details.

Recommendation

Reject

---

## Editorial Decision

awaiting_resubmission